# Determination of Floral Origin Markers of Latvian Honey by Using IRMS, UHPLC-HRMS, and ^1^H-NMR

**DOI:** 10.3390/foods11010042

**Published:** 2021-12-24

**Authors:** Kriss Davids Labsvards, Vita Rudovica, Rihards Kluga, Janis Rusko, Lauma Busa, Maris Bertins, Ineta Eglite, Jevgenija Naumenko, Marina Salajeva, Arturs Viksna

**Affiliations:** 1Department of Chemistry, University of Latvia, Jelgavas Street 1, LV-1004 Riga, Latvia; vita.rudovica@lu.lv (V.R.); rihards.kluga@lu.lv (R.K.); janis.rusko@bior.lv (J.R.); lauma.busa@lu.lv (L.B.); maris.bertins@lu.lv (M.B.); jn18020@students.lu.lv (J.N.); ms18103@students.lu.lv (M.S.); arturs.viksna@lu.lv (A.V.); 2Institute of Food Safety, Animal Health and Environment “BIOR”, Lejupes Street 3, LV-1076 Riga, Latvia; 3Latvian Beekeeping Association, Rigas Street 22, LV-3004 Jelgava, Latvia; ineta.eglite@strops.lv

**Keywords:** honey, light stable isotope mass spectrometry, ultra-high performance liquid chromatography, high resolution mass spectrometry, nuclear magnetic resonance, principal component analysis, floral origins

## Abstract

The economic significance of honey production is crucial; therefore, modern and efficient methods of authentication are needed. During the last decade, various data processing methods and a combination of several instrumental methods have been increasingly used in food analysis. In this study, the chemical composition of monofloral buckwheat (*Fagopyrum esculentum*), clover (*Trifolium repens*), heather (*Calluna vulgaris*), linden (*Tilia cordata*), rapeseed (*Brassica napus*), willow (*Salix cinerea*), and polyfloral honey samples of Latvian origin were investigated using several instrumental analysis methods. The data from light stable isotope ratio mass spectrometry (IRMS), ultra-high performance liquid chromatography coupled with high-resolution mass spectrometry (UHPLC-HRMS), and nuclear magnetic resonance (NMR) analysis methods were used in combination with multivariate analysis to characterize honey samples originating from Latvia. Results were processed using the principal component analysis (PCA) to study the potential possibilities of evaluating the differences between honey of different floral origins. The results indicate the possibility of strong differentiation of heather and buckwheat honeys, and minor differentiation of linden honey from polyfloral honey types. The main indicators include depleted δ^15^N values for heather honey protein, elevated concentration levels of rutin for buckwheat honey, and qualitative presence of specific biomarkers within NMR for linden honey.

## 1. Introduction

Due to its sweet taste and antibacterial properties, honey is in high demand in today’s market. In 2018, approximately 2000 tons of honey from 103,000 beehives were produced in Latvia. In Europe more generally, the demand for honey is higher than local producers can produce, and therefore a large part is imported [1]. Honey is an expensive product when compared to other sweeteners. Counterfeit honey is considered to be honey that contains added amounts of other cheaper sweeteners. Directive 2014/63/EU of the European Parliament and the Council clearly defines what constitutes natural honey. To protect the interests of consumers and regulate the fair price of honey in today’s market, methods of honey authenticity and quality indicators are constantly evolving. One or several modern instrumental methods are increasingly used with which quality characteristics are determined, as well as the authenticity of honey is assessed by applying chemometric methods [2].

Although IRMS has been used mainly to determine the presence of C_4_ sugar additives, measurements of light stable isotopes have increasingly been used to determine the botanical or geographical origin of honey. δ^13^C and δ^15^N values for honey and proteins provide useful information in distinguishing samples of different floral types of honey (acacia, chestnut, citrus, eucalyptus, rhododendron, and polyfloral honey) [3].

Polyphenol compound concentrations are considered as potential useful variables for floral origins [4]. The polyphenol profile is a useful tool for geographical and floral origin assessment. A robust UHPLC-HRMS method for polyphenol quantification is often used [5,6,7]. Sugaring-out assisted liquid–liquid extraction (SULLE) sample preparation has been proven to be an optimal choice of honey studies using HRMS equipment [5].

Nuclear magnetic resonance (NMR) is increasingly used to evaluate the authenticity of honey. The information provided by the proton NMR spectrum, in combination with various chemometric methods, is used to distinguish between honey of different botanical origins. Depending on the task to be performed, chemometrics can be performed for the whole spectrum or only for a certain interval. In most cases, the region characteristic of aromatic compounds (9–6 ppm) or the region characteristic of aliphatic compounds (3–0.5 ppm) are used [8,9]. A study in Brazil successfully distinguished between citrus, eucalyptus, and wildflower honey, and some honey was found to be counterfeit [10]. A similar approach was used by Spiteri et al. for the assessment of geographical origin [11]. Samples of acacia honey from Eastern Europe and Italy were compared. Due to the different flora, characteristic floral markers were found in the samples [12].

Principal component analysis (PCA) in chemistry allows for the study of the properties of different datasets of chemical compounds. Determining which compounds have similar properties and which study objects form groups, one can also try to predict the properties of the study object or belonging to a group. Various instrumental analyses are practically effective for the analysis of principal components, wherein the spectral image is obtained, for example, the total ion chromatogram after the retention time, under different conditions [13]. Quantitative values of various honey compounds, isotope ratio values, etc., quality indicators can be used to analyze the principal components. Depending on the purpose of the study (counterfeits, origin of flowers, geographical regions, etc.), honey types are selected, chemical instrumental analyses are performed, and the results are used for the analysis of principal components to determine the formation of groups [10,14,15,16].

The main aim of this study was the use of different methodologies to classify the botanical origin of various types of monofloral Latvian honey to target the mislabeling of protected destination of origin (PDO) products. One of the goals was to gather the data on fresh samples collected directly from the beekeepers of Latvia instead of processed and commercially available honey. Further, we validated the true floral origin using melissopalynology analysis. Finally, we evaluated multiple criteria to classify individual monofloral variety honeys by using a combination of analytical methods (IRMS, UHPLC-HRMS and NMR) and statistical treatment of experimental data and PCA analysis.

## 2. Materials and Methods

### 2.1. Samples

A total of 78 different honey samples were collected directly from the beekeepers in the territory of Latvia, declared as of natural origin and of specific monofloral varieties. The true botanical origin of the samples was further examined by melissopalynology analysis [17] and later confirmed or deemed of lesser, polyfloral quality. The criteria of specific pollen for monofloral honey [18] was reached for 4 buckwheat (*Fagopyrum esculentum*) (>25%), 6 clover (*Trifolium repens*) (>45%), 3 heather (*Calluna vulgaris*) (>40%), 3 linden (*Tilia cordata*) (>17%), 4 rapeseed (*Brassica napus*) (>70%), and 3 willow (*Salix cinerea*) (>45%) honey samples. The other 55 honey samples were polyflorals and kept for honey analysis to make an assessment for the capability of potential floral origins indicators.

### 2.2. IRMS

#### 2.2.1. Protein Extraction by Dialysis

Honey proteins were prepared according to the method described by Bilikova [19] with slight readjustments. A 15 g sample of honey was weighed, and 15 mL of deionized water was added. Semi-permeable SnakeSkin (10K MWCO) dialysis membrane was filled with a homogeneous clear honey solution. After dialysis, the purified protein solution was quantitatively transferred into a beaker and placed in the drying oven at 40 °C for about 48 h until the proteins were dried. Then, proteins were weighed and stored at 4 °C until IRMS analysis.

#### 2.2.2. δ^13^C and δ^15^N, and Total Carbon and Nitrogen Analysis

Continuous flow IRMS (Nu Horizon) coupled with an elemental analyzer (EuroEA3024) was used for the analysis. The complete combustion of the samples and the operation of the element analyzer were verified by performing stability tests on the equipment. Certified reference materials USGS-40 and USGS-41 were used as reference materials. The device conditions were prepared as described in previously published method [20].

### 2.3. UHPLC-HRMS

#### 2.3.1. Chemicals

Analytical standards of 3,4-dihydroxybenzoic acid (>98.2%), acacetin (>98.7%), apigenin (>99%), caffeic acid (>98.5%), catechin (>99%), chlorogenic acid (>99%), chrysin (>99%), daidzein (>99%), galangin (>98.5%), gallic acid (>95.5%), genistein (>99%), (-)-epicatechin (>90.3%), folic acid (>91.2%), formononetin (>99%), isovitexin (>99%), luteolin (>99.9%), myricetin (>98%), o-coumaric acid (>99.7%), p-coumaric acid (>99.6%), p-hydroxybenzoic acid (>99.9%), pantothenic acid (>98.6%), phenylacetic acid (>99.7%), rhamnetin (>99%), rutin trihydrate (>94%), quercetin (>98%), sinapic acid (>96%), syringic acid (>98.5%), trans-ferulic acid (>99.8%), vanillic acid (>98.2%) were purchased from Extrasynthese (Lyon, France) or Sigma-Aldrich (St. Louis, MO, USA). The standard of (-)-cis, trans-abscisic acid (>99.9%) was purchased from Santa Cruz Biotechnology (Dallas, TX, USA), and kaempferol (>97%) was purchased from ChromaDex (Santa Ana, CA, USA). HPLC-MS grade acetonitrile (MeCN) and dimethyl sulfoxide (DMSO) were purchased from Merck (Darmstadt, Germany), while formic acid (FA), hydrochloric acid (HCl), and sodium chloride (NaCl) were purchased from Sigma-Aldrich.

#### 2.3.2. SULLE Sample Preparation

Samples were prepared by the previously published SULLE method [5]. A total of 0.5 g of honey was added in an Eppendorf tube within 0.5 mL of 10% NaCl in 0.01M HCl (pH = 2). A total of 1 mL of MeCN was added to the mixture, and the tube was vortexed for another 1 min at 2000 rpm followed by 1 min centrifugation at 15,000 rpm. The upper organic phase was collected in a 2 mL crimp top chromatography vial. The procedure was repeated until the total collected organic phase amount of about 1.9 mL. The organic phase was dried under a gentle nitrogen flow at room temperature and reconstituted in 0.5 mL of water/MeCN mixture (98:2 *v*/*v*) with added 0.1% FA. Extracts were stored at 4 °C, in the dark, before the analysis.

#### 2.3.3. UHPLC-HRMS Systems

Liquid chromatography analysis was performed using a Dionex UltiMate 3000 UHPLC system (Thermo Scientific, Oleten, Switzerland) equipped with a Kinetex PFP column (3.00 × 100 mm, 1.7 μm, 100 Å), obtained from Phenomenex (Torrance, CA, USA). LC system was coupled to a high-resolution mass spectrometer Q Exactive (Thermo Scientific, Bremen, Germany). LC parameters: 5 μL injection volume, 40 °C column temperature, 10 °C sample temperature, flow rate set to 0.450 mL·min^−1^, diverter valve was switched to mass spectrometer at 1.3 min. The mobile phase A (0.1% formic acid in H_2_O) and B (0.1% formic acid in MeCN) were used in gradient mode: 4 min preinjection equilibration held at 2% B; 0–3 min at 2–5% B; 3–9 min at 5–98% B; 9–13 hold at 98% B; 13–14 return to the initial 2% B.

Heated electrospray (HESI) interface was used in positive and negative ionization mode, and polarity switching method was used with the following parameters: ion source voltage in negative/positive ionization (2500 V/3500 V), 280 °C temperature for ion transfer tube, 450 °C evaporator temperature.

### 2.4. NMR

#### 2.4.1. Sample Preparation

The method proposed by Schievano et al. was used and adjusted for available equipment to acquire ^1^H-NMR spectra of honey [21]. A total of 200 ± 3 mg of honey was dissolved in 1.0 mL of D_2_O buffer solution. The resulting solution was transferred to an NMR tube, and ^1^H-NMR spectra were acquired. D_2_O buffer solution was prepared by dissolving 1.02 g of KH_2_PO_4_ and 0.96 mg of NaN_3_ in 20 mL of D_2_O. The buffer solution pH was adjusted to 4.4 with 85% H_3_PO_4_.

#### 2.4.2. ^1^H-NMR Spectra Acquisition

NMR spectra were acquired with Bruker BioSpin GmbH, Rheinstetten, Germany, Fourier300 spectrometer (working frequency of 300 MHz for ^1^H) equipped with a 5 mm DUL 13C-1H/D Z-gradient EasyProbe. ^1^H-NMR spectra were acquired with noesypr1d pulse program using 125 ms mixing time and −40 dBW presaturation power level during recycle delay and mixing time, 2 s relaxation delay (D1), 6103 Hz spectral width, 64k points of time-domain (TD), and 8 dummy scans (DS). The acquisition time for one scan was 5.37 s. Constant receiver gain (rg = 3) was used.

#### 2.4.3. ^1^H-NMR Spectra Processing

Acquired ^1^H-NMR spectra were processed with MestReNova software (version 14.1.1). FID was zero-filled to 128k points, and exponential apodization (0.3 Hz) was used. Manual phase correction and automatic baseline correction (Whittaker smoother) were performed. Chemical shifts were referenced to α-glucopyranose doublet (δ = 5.320 ppm). ^1^H spectra were binned using signal integral sum 0.5–3.0, 6.0–9.0 ppm with a bin width of 0.01 ppm. The binned spectra were normalized to the total area.

### 2.5. Statistical Analysis

The data processing was performed using statistical software Minitab 17.1.0. One-way ANOVA analysis of variance was performed in order to assess the significant differences of the variable between monofloral and polyfloral honey samples. The confidence level (*p* = 0.05) was used for every ANOVA test. Tukey comparison procedure for assuming equal variances was used for every variable obtained from IRMS and UHPLC-HRMS methods while Fisher comparison was used once for total N assessment in honey proteins. Principal component analysis was performed for data reduction in order to find potential chemical compound biomarkers for floral origins. The correlation matrix was used for analysis. As a pre-step, the software performed standardization of variables, meaning a variable was rescaled to have a mean of zero and a standard deviation of one. Principal component scores and their correlation coefficients are stored in a Appendix A. The formation of monofloral group clusters or positions in the score plot was used to assess the potential of variable capability as a marker.

## 3. Results and Discussion

### 3.1. IRMS Analysis of Honey Proteins

C and N isotope ratio and total weight fraction of monofloral and polyfloral honey samples is presented in Table 1. The ANOVA one-way results show that there was no significant variance, with a confidence level of 95% between monofloral and polyfloral honey, by using δ^13^C values (*p* = 0.08). Tukey test simultaneous differences of means for δ^13^C are described in Appendix A.

The carbon isotope ratio in honey proteins is directly influenced by carbon fixation in plants from which bees are gathering honey. Therefore, carbon isotope analysis is mainly used for C4 plant additive determination in honey. Nevertheless, the δ^13^C values are often used for floral origins determination [3,22].

After extractions of sugars, the honey proteins showed δ^13^C values in a range of −25.47‰ to −29.64‰, which is characteristic for C_3_ plants [23]. Moreover, in comparing polyfloral honey proteins (δ^13^C = −27.4 ± 0.9‰) with other results, we found that δ^13^C values are more depleted than of Mediterranean region honey proteins. δ^13^C values are dependent on the amount of sun exposure to plants and air humidity; therefore, an increase of sunny days and less precipitations increases the δ^13^C values [24].

The nitrogen isotope ratio for honey proteins reflects the nitrogen content of the soil where plants from which bees have gathered the nectar grow. δ^15^N > 0.0‰ values indicate that the nitrogen is biologically fixed in soil, and values near 0.0‰ show that the nitrogen is obtained from air. Results show clover honey proteins are enriched with heavy nitrogen isotope, although the plant is considered as gathering nitrogen via *Rhizobium* bacteria [25]. Exceptional honey proteins were extracted from heather honey, indicating depleted nitrogen ratio values and statistically different significance (*p* = 0.001) using ANOVA one-way Tukey tests (see Appendix A). In total, 11 out of 78 honey proteins showed negative δ^15^N values. These samples of honey were heather monoflorals and polyflorals that had reported the presence of heather (*Calluna vulgaris*) pollen (see the Appendix A).

Total carbon and nitrogen in honey proteins were found to have no particular statistical difference using the ANOVA test. *p*-values were found for total carbon (*p* = 0.5) and total nitrogen (*p* = 0.06), although total nitrogen *p*-values were close to 0.05, which suggests that results could be capable for further floral origin discrimination investigation. Using the Fisher test, we found that there are differences in heather and buckwheat (increased total nitrogen) honey proteins between willow and rapeseed (decreased total nitrogen) (see Appendix A). Total nitrogen in proteins generally is ≈16% [26]. Obtained nitrogen mass fraction results suggest that after dialysis, pure protein is not obtained, but instead a mixture of protein and other molecularly large compounds that could not be separated via dialysis such as lipids and pollen [27,28].

### 3.2. UHPLC-HRMS Analysis

A total of 31 organic compounds (13 phenolic acids, 14 flavonoids, 2 vitamins, 2 plant hormones) were successfully quantified in polyfloral honey and various monoflorals. The biochanin A, biotin, and procyanidin A2 were found only in a few samples near the LOQ, and these compounds were omitted for future assessments.

In Figure 1, concentrations for the 27 most common found compounds in polyfloral honey of origins of Latvia are shown. The highest concentrations of phenolic acids were obtained for p-hydroxybenzoic acid (3923 ± 3522 μg/kg), abscisic acid (4174 ± 2238 μg/kg), p-coumaric acid (2685 ± 1271 μg/kg), and ferulic acid (1638 ± 572 μg/kg) while kaempferol (1432 ± 728 μg/kg) was the flavonoid and pantothenic acid B5 (986 ± 412 μg/kg) was vitamin with the highest average concentrations.

Formononetin, chrysin, and folic acid were not shown by boxplot because these compounds were found over LOQ only 6 to 21 polyflorals, suggesting these compounds are characteristic of a specific floral origin. Apigenin was also omitted, although it was found in 43 polyfloral honey samples but slightly over LOQ, and the mean value was 2 ± 3 μg/kg.

The results come in good agreement with another study showing similar concentration levels of the same compounds, except apigenin, which was found in larger concentrations by Lo Dico et al. [7]. The one-way ANOVA tests of Tukey comparison were performed to honey groups of different floral origins. The six compounds showed statistically significant differences that could be used for monofloral honey samples or speciation of honey floral origins. Rutin interval plot and graphical summary of differences of mean are shown in Figure 2.

Rutin showed a statistically significant difference in buckwheat honey and in the other types of honey. In buckwheat honey, rutin showed a concentration of 572 ± 167 μg/kg, while polyfloral honey contained from <5 (LOQ) to 696 μg/kg with a mean of 53 μg/kg. Two polyfloral samples (P5 and P42) had notably higher concentrations of rutin, corresponding to 649 and 696 μg/kg, respectively, equivalent to high buckwheat pollen presence for polyflorals (17 and 24%, respectively). It was less found in linden and rapeseed honey and not found at all in heather honey. This comes in good agreement with melissopalynology results, as buckwheat (*Fagopyrum esculentum*) pollen was found in clover and willow monofloral honeys in a range of 0–6%. Other statistically significant differences within honey floral origins were found using vanillic acid, quercetin, p-hydroxybenzoic acid, p-coumaric acid, and pantothenic acid B5 (see Table 2). The monofloral clover and willow honey interfered to discriminate buckwheat honey from other types of floral origins using p-hydroxybenzoic acid and p-coumaric acid concentrations. Interference could be explained by buckwheat pollen presence in clover and willow monofloral honey. The quercetin concentrations showed a statistical difference between buckwheat and heather honey. While quercetin has no potential as a specific floral marker, it would be very helpful, since both share similar visual properties as dark-colored honeys [29,30]. Similarly, pantothenic acid and vanillic acid can be used for specific floral origin request determination, or could be a helpful indicator with a combination of other variables.

Comparing honey of Polish origins, the authors of [31] found similar levels of p-coumaric acid and quercetin in heather honey. However, Latvian honey showed lower concentrations of chrysin, galangin, and apigenin but higher concentrations of luteolin than Polish honey. Rapeseed honey of Romanian origins share similar levels of chlorogenic acid and p-coumaric acid but increased of gallic acid, p-hydroxybenzoic acid, 3,4-dihydroxybenzoic acid, vanillic acid, caffeic acid, and myricetin [32]. In another study, p-hydroxybenzoic acid is mentioned as a commonly found compound in clover and heather honey. Moreover, p-coumaric and vanillic acid are reported as commonly found in heather honey, while our study shows that concentration levels were not different from Latvian polyfloral honey. Quercetin is usually found in clover honey [33]. Regardless of other studies, recent preliminary UHPLC-HRMS results of Latvian honey showed rutin as a suggestable indicator for buckwheat honey. However, increased rutin concentration levels for a few polyflorals containing notable buckwheat pollen percentage were also observed. This suggests a need for further investigation to determine a threshold level of rutin in order to distinguish buckwheat honey from polyfloral honeys.

### 3.3. Principal Component Analysis

All data for PC scores and loadings are available in the Appendix A in the form of excel spreadsheets. The carbon and nitrogen isotope ratio and total element percentage were used for PCA to determine the Latvian honey floral origins using a single IRMS method. The variables were standardized, and a correlation matrix was used since variables were expressed in different units of measurement. Eigenvalues were expressed in the scree plot (see Appendix A). PC1-PC3 described variability by 94.6%, and these components were used for evaluation. A total of 15 samples were considered as outliners using a Mahalanobis distance criteria and were withdrawn from PCA. The outliner samples were coded as follows: monofloral buckwheat (B1), clover (C1), rapeseed (R2), willow (W1), and polyfloral (P2, P5, P8, P18, P20, P23, P24, P26, P33, P42, P47) honey. In Figure 3a, the heather honey formed a cluster away from other honey samples because of PC1. After the investigation of loading coefficients (see Figure 3b), it appears that PC1 had a high positive correlation (r = 0.50) of nitrogen isotope ratio and total carbon in proteins (r = 0.53) but negative correlation of total nitrogen in proteins (r = −0.67). The monoflorals of heather honey were significantly different (*p* = 0.001) of depleted δ^15^N values, while total carbon and total nitrogen showed no significant differences. Nevertheless, in comparing the means of the heather honey and other types of origins, we found that the mean of total carbon was the lowest, and total nitrogen was highest for heather honey. Polyflorals (P3, P6, P14, P55) that formed cluster with monofloral heather honey also contained heather pollen (31%, 10%, 38%, 4%). The honey sample P46 had 6% of heather pollen content and it was the only honey sample that had heather pollen more than 4%; moreover, it was not located in the cluster. Other polyfloral honey samples with heather pollen >4% (P2—24%; P8—35%; P18—22%) were classified as outliners and had similar δ^15^N values but increased means of total carbon and decreased means of total nitrogen. This highlights the need to monitor the total carbon and nitrogen content in honey protein IRMS analysis when monofloral heather honey purity must be assessed.

The concentrations of phenolic acids, flavonoids, vitamins, and plant hormones in honey were used for PCA of UHPLC-HRMS assessment. The catechin, chrysin, folic acid, and formononetin were omitted for evaluation, and 27 compound concentrations (μg/kg) were standardized and a correlation matrix was constructed. After evaluation of the scree plot (see Appendix A), we used the PC1-PC3 for further analysis, since they cover the most variability of data (45.7%). A total of 11 samples were considered as outliners using a Mahalanobis distance criteria and were withdrawn from PCA. The outliner samples were coded as follows: monofloral clover (C2, C5) and polyfloral (P7, P8, P20, P26, P29, P37, P43, P46, P53) honey. The formation of an exceptional cluster such as with IRMS results were not observed. The PC2 has strong positive correlation of p-hydroxybenzoic acid (r = 0.39), rutin (r = 0.38), and p-coumaric acid (r = 0.34), and these compounds were previously discussed as potential buckwheat honey floral markers. The location of buckwheat honey in score plot (see Figure 4a) was outside of the majority of samples, but buckwheat honey was found to have higher PC2 scores than other honey. The polyflorals (P5, P42, P51, P54) with similar PC2 scores also had buckwheat pollen (17%, 24%, 16%, 4%). The PC3 was not selective for certain floral group but depended on the ratio of 3,4-dihydroxybenzoic acid (r = 0.43) and abscisic acid (r = −0.35) concentrations.

Due to the presence of a wide range of chemical compounds found in honey, NMR is considered to be one of the most prominent methods for food analysis [34]. The complete identification of chemical compounds from ^1^H-NMR spectra is a difficult task because of compound low concentrations and signal overlays. Nevertheless, the honey of similar floral origins share a similar ^1^H-NMR spectra image, and therefore principal component analysis (PCA) was used to conduct an assessment of honey without full quality analysis. The ^1^H-NMR spectra of honey samples were transformed into the spectral bins from 0.5 to 3 ppm (aliphatic region) and 6 to 9 ppm (aromatic region), with a bin width of 0.01 ppm before the principal component analysis. Carbohydrate region (3–6 ppm) was excluded for the PCA due to the presence of high intensity peaks that are sensitive to scaling method prior to PCA and strongly affect cluster forming in a PCA plot [35]. Furthermore, minor and specific carbohydrate ^1^H NMR signals were not resolved using 300 MHz NMR spectrometer, which could improve PCA discrimination [21,36,37]. The obtained scree plot and PCA plots of Latvian honey samples are shown in Appendix A. The PC1-PC3 had exceptionally high contribution of data variability, explaining 29.2% of the variance from 552 variables. PCA plot with PC1 and PC3 of studied Latvian honeys of monofloral honeys (Figure 5a) could be described in several groups as follows: (1) buckwheat, clover, and willow honeys with mostly negative PC1; (2) linden honeys with positive PC1 and mostly positive PC3; and (3) heather honeys with positive PC1 and negative PC3. Rapeseed honeys showed cluster near PC1 and PC3 cross-point that indicated absence of specific compounds. Honey grouping could be explained by using PC1 and PC3 loading plots (see Figure 5b,c). In the case of buckwheat, clover, and willow honeys, ^1^H-NMR spectral bins with δ = 6.87–6.82, 7.16–7.21, 1.67–1.74, and 0.90–1.02 ppm contributed to negative PC1 score. This can be explained with the presence of tyrosine (δ = 6.87–6.82 and 7.16–7.21 ppm), leucine (δ = 1.67–1.74 ppm), and isoleucine and valine (δ = 0.90–1.02 ppm). These amino acids have been previously found in a higher level for buckwheat honey [38]. Surprisingly, in Latvian monofloral clover and willow honeys, these amino acids were found as well. For the linden monofloral honey, ^1^H-NMR spectral bins with δ = 2.40–2.47 and 7.15–7.23 ppm contributed to positive PC1 score, and bins with δ = 6.14–6.18 and 7.15–7.23 ppm for positive PC3 score. Linden honey ^1^H-NMR spectra-specific bins can be attributed to the cyclohexa-1,3-diene-1-carboxylic acid (CDCA) derivatives (δ = 6.14–6.18 and 7.15–7.23 ppm) that are specific markers of monofloral linden honey [38]. Lastly, the heather honey showed resolved cluster position in PCA plot that was mostly affected by ^1^H-NMR spectral bins with δ = 7.28–7.32 and 2.37 ppm. These bins can be assigned to the previously found carboxylic acids, such as phenylacetic acid, 3-phenyllactic acid and benzoic acid (δ = 7.28–7.32 ppm), and pyruvic acid (δ = 2.37 ppm) [38,39]. Typical binned ^1^H-NMR spectra of analyzed monofloral honeys with the assigned compounds are shown in Appendix A. It was shown that PCA in combination with ^1^H-NMR showed clear separation of monofloral heather honey from other studied honeys. Unfortunately, monofloral honeys with negative PC1 could not be resolved in separate clusters, and other statistical methods should thus be used (e.g., OPLS-DA) [38,40].

## 4. Conclusions

The chemical profile of monofloral buckwheat, clover, heather, linden, rapeseed, willow, and polyfloral honey samples of Latvian origins was assessed by IRMS, UHPLC-HRMS, and NMR methods in order to find suitable indicators that could be used for the classification of botanical origin of honey. The depletion in δ^15^N values in honey proteins was suggested as indicator for heather honey (δ^15^N = −2.3 ± 1.0‰). Moreover, the total N in proteins indicated potential distinctiveness between the pairs of willow and rapeseed honey, and buckwheat and heather honey. After the data treatment using PCA, the total nitrogen and total carbon in honey proteins were taken into account for recognition of heather honey origins. Out of 31 organic compounds quantified by UHPLC-HRMS, rutin showed a selective difference as a buckwheat honey indicator. p-Hydroxybenzoic acid, p-coumaric acid, pantothenic acid (B5), quercetin, and vanillic acid were found to have statistically different concentration levels within different monofloral honey types and could be used as specific indicators for monofloral honey purity. The polyphenol profile comes in good agreement with other studies, with the exception of the few compounds that were reported with higher concentrations in foreign country honey. The NMR qualitative analysis showed distinguishment among monofloral buckwheat, heather, and linden honey. Using NMR tyrosine, proline, alanine, and lactic acid, we found characteristic chemical shifts in buckwheat honey, with monosubstituted benzene derivatives and ethanol in heather honey and CDCA derivatives in linden honey. This study proves the validity of the combination of multiple analytical methods, statistical data treatment, and PCA to differentiate various natural monofloral honey classes, thus guaranteeing botanical authentication and the honey quality and origin.

## Figures and Tables

**Figure 1 foods-11-00042-f001:**
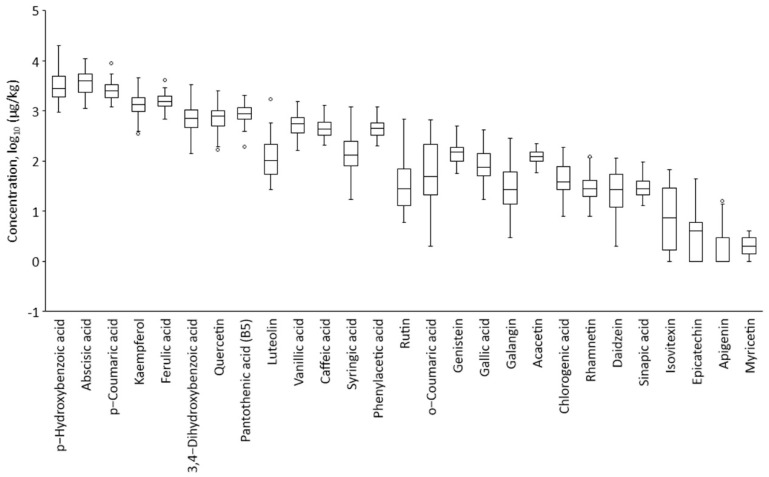
Boxplots of organic compound concentration (μg/kg) in polyfloral honey determined by UHPLC-HRMS; results converted in decimal logarithm scale.

**Figure 2 foods-11-00042-f002:**
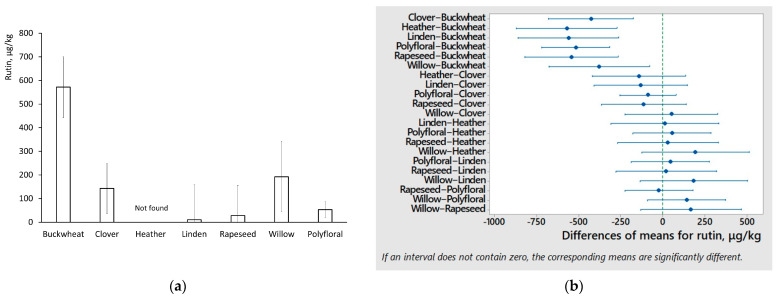
One-way ANOVA test of statistically significant difference between monofloral buckwheat, clover, heather, linden, rapeseed, willow, and polyfloral honey (**a**) using interval plot (μg/kg) as graphical summary with 95% confidence interval bars, and (**b**) using Tukey comparison of 95% confidence intervals.

**Figure 3 foods-11-00042-f003:**
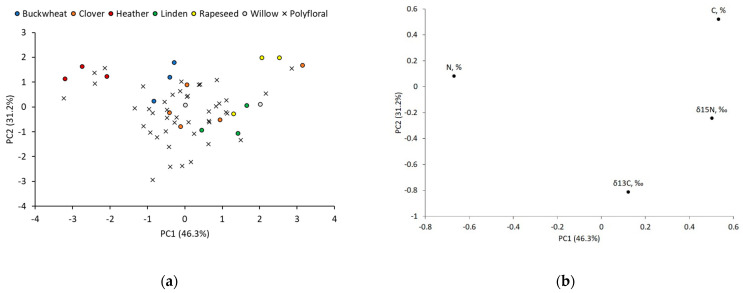
PCA of monofloral buckwheat, clover, heather, linden, rapeseed, willow, and polyfloral honey samples: (**a**) score plot between PC1 and PC2, and (**b**) loading plot of variables obtained by IRMS (δ^13^C, δ^15^N, total C and N in honey proteins).

**Figure 4 foods-11-00042-f004:**
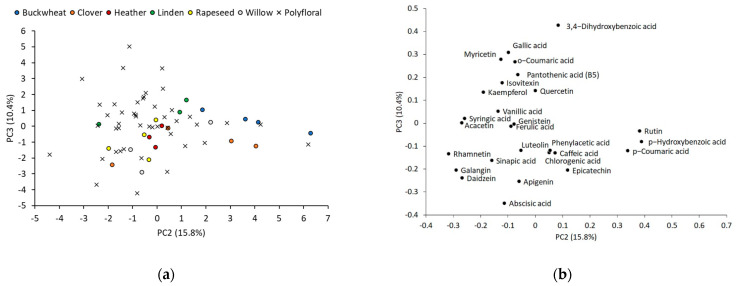
PCA of monofloral buckwheat, clover, heather, linden, rapeseed, willow, and polyfloral honey samples: (**a**) score plot between PC2 and PC3, and (**b**) loading plot of 27 organic compound concentrations obtained by UHPLC-HRMS.

**Figure 5 foods-11-00042-f005:**
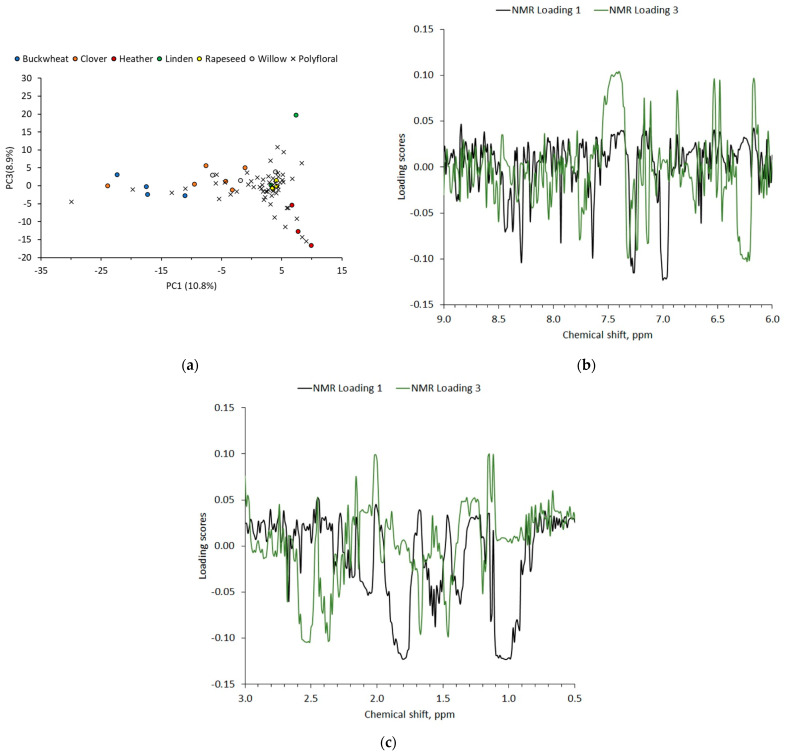
PCA of monofloral buckwheat, clover, heather, linden, rapeseed, willow, and polyfloral honey samples: (**a**) score plot between PC1 and PC3; loading plots of PC1 and PC3 for ^1^H-NMR spectra with bin width 0.01 ppm for intervals (**b**) 9.0–6.0 ppm and (**c**) 3.0–0.5 ppm.

**Table 1 foods-11-00042-t001:** IRMS analyses results by floral origins of honey.

Floral Origins	δ^13^C, ‰		δ^15^N, ‰		Total C, %		Total N, %		
	Mean	SD	Mean	SD	Mean	SD	Mean	SD	N
Buckwheat	−28.7	0.7	6.8	1.5	48.1	1.4	9.6	0.5	4
Clover	−27.7	0.9	6.5	1.7	50	4	8	2	6
Heather	−28.13	0.10	−2.3	1.0	47.4	0.6	10.0	0.6	3
Linden	−26.7	0.2	5.8	0.7	50	2	7.0	0.7	3
Rapeseed	−27.5	0.5	4.9	1.1	53	4	6.4	0.8	4
Willow	−27.6	0.5	6.5	1.0	56	7	6	2	3
Polyfloral	−27.4	0.9	4	3	51	6	8	2	55

**Table 2 foods-11-00042-t002:** Comparison of chemical compound mass concentrations (μg/kg) quantified by UHPLC-HRMS that share statistically significant differences between groups of floral origins.

Floral Origins	Mean ± SD, μg/kg				
	p-Hydroxybenzoic Acid	p-Coumaric Acid	Pantothenic Acid (B5)	Quercetin	Vanillic Acid
Buckwheat	13,863 ± 4472 ^A^	5561 ± 1159 ^A^	910 ± 247 ^AB^	1297 ± 511 ^A^	602 ± 329 ^AB^
Clover	7907 ± 4809 ^AB^	3963 ± 991 ^AB^	764 ± 193 ^AB^	523 ± 204 ^AB^	477 ± 164 ^AB^
Heather	2984 ± 494 ^B^	2519 ± 738 ^B^	1513 ± 250 ^A^	198 ± 86 ^B^	190 ± 29 ^B^
Linden	1423 ± 1004 ^B^	2509 ± 161 ^B^	558 ± 243 ^B^	475 ± 390 ^AB^	447 ± 337 ^AB^
Rapeseed	1740 ± 248 ^B^	2341 ± 499 ^B^	577 ± 87 ^B^	986 ± 167 ^AB^	725 ± 28 ^AB^
Willow	6753 ± 3252 ^AB^	4550 ± 1529 ^AB^	1017 ± 131 ^AB^	726 ± 445 ^AB^	1014 ± 619 ^A^
Polyfloral	3923 ± 3522 ^B^	2685 ± 1271 ^B^	986 ± 412 ^AB^	824 ± 419 ^AB^	585 ± 288 ^AB^

^AB^—results marked with a different superscript letter are significantly different using ANOVA one-way Tukey test (*p* < 0.05). Letter “A” indicates affiliation to a group with higher means and letter “B” indicates affiliation to a group with lower means, while “AB” shows affiliation for both groups.

## Data Availability

Data are contained within the article or Appendix A.

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
