# Peer review of "Determination of Floral Origin Markers of Latvian Honey by Using IRMS, UHPLC-HRMS, and 1H-NMR"

_foods, 2021, doi:10.3390/foods11010042_

Round 1
Reviewer 1 Report
Overall the paper needs amendments before is considered for publication.
The novelty of the study needs to be addressed properly. Background needs improvements as aim is not justified properly.
Abstract to be re- written and include more information on the outcomes of the research.
More critical evaluation to be included in discussion section.
Conclusion is response of the aim/ objectives. Please amend.
Methodology needs amendments. Some parts are too extensive while some lack proper explanation and reference support. Some of the references are too old e.g. 1991.
Author Response
REVIEWER 1
Comments and Suggestions for Authors
- Overall the paper needs amendments before is considered for publication.
Response:
Authors highly appreciate the input of the first review cycle, major changes and additions have been implemented following the recommendations suggested by Reviewers 1 & 2. We believe the quality of the manuscript has improved significantly and we hope that reviewers also think the same.
- The novelty of the study needs to be addressed properly. Background needs improvements as aim is not justified properly.
Response:
We appreciate the reviewer's suggestion for improvements to this essential part of the article. We have concisely addressed the novelty and main aim of the manuscript within the final paragraph of introduction. We feel that now it is much more unambiguous.
It now reads as follows:
“The main aim of this study was to investigate a methodology to classify the botanical origin of various types of monofloral Latvian honey to target the mislabeling of protected destination of origin (PDO) products. One of the goals was to gather the data on fresh samples collected directly from the beekeepers of Latvia instead of processed and commercially available honey. Further, we validated the true floral origin using melissopalynology analysis. Finally, we evaluated multiple criteria to classify individual monofloral variety honeys by using a combination of analytical methods (IRMS, UHPLC-HRMS and NMR) and statistical treatment of experimental data and PCA analysis.”
- Abstract to be re-written and include more information on the outcomes of the research.
Response:
Authors appreciate the reviewer's suggestion and it has been re-written to highlight the results better. Additional sentences have been added and the text has been reorganised.
The final lines of abstract now read as follows:
“Results were processed using the principal component analysis (PCA) to study the potential possibilities to evaluate the differences between honey of different floral origins. The results indicate strong differentiation of heather and buckwheat honeys, and minor differentiation of linden honey from polyfloral honey types. The main indicators include: depleted δ15N values for heather honey protein, elevated concentration levels of rutin for buckwheat honey and qualitative presence of specific biomarkers within NMR for linden honey.”
- More critical evaluation to be included in discussion section.
Response:
We acknowledged the reviewer's inquiry and feel that the discussion section has benefited from the changes made. Some changes also have been made in part as a response to questions raised by the reviewer #2. Briefly, we would like to introduce the changes.
In section 3.2. UHPLC-HRMS analysis, we have introduced the following text (lines 256-260):
“In buckwheat honey, rutin showed concentration of 572 ± 167 μg/kg while polyfloral honey contained from < 5 (LOQ) to 696 μg/kg with a mean of 53 μg/kg. Two polyfloral samples (P5 and P42) had notably higher concentrations of rutin, corresponding 649 and 696 μg/kg, corresponding to high buckwheat pollen presence for polyflorals (17 and 24%).”
Addition of lines 287-292:
“Regardless of other studies, recent preliminary UHPLC-HRMS results of Latvian honey show rutin as a suggestable indicator for buckwheat honey. Although, increased rutin concentration levels for few polyflorals containing notable buckwheat pollen percentage are observed too. This suggests a need for further investigation to determine a threshold level of rutin in order to distinguish buckwheat honey from polyfloral honeys.”
Addition of lines in section 3.3. Principal component analysis, to compile results explained previously in paragraph (lines 318-320):
“This highlights the need to monitor the total carbon and nitrogen content in honey protein IRMS analysis when monofloral heather honey purity must be assessed.”
Major changes have been done in the context of the previous work section “Qualitative NMR analysis”. The section itself has been discarded and the most relevant results transferred to section 3.3. Principal component analysis. The 1H-NMR spectra comparison of all monofloral honey types was made and put in Supplementary material. Following changes are rewritten from line 344 to 385. Additionally, Figure 5A, 5B, 5C have been edited to better track results.
- Conclusion is response of the aim/ objectives. Please amend.
Response:
The authors thank the reviewer for the notice and per request we have amended and changed the conclusion section to better fit the manuscript aims and objectives and to conclude the manuscript. A large part of the conclusion section had already highlighted the most significant findings of the research project in order to make the reading easier. Please find the changes in manuscript text and we hope they are sufficient.
- Methodology needs amendments. Some parts are too extensive while some lack proper explanation and reference support. Some of the references are too old e.g. 1991.
Response:
The methodology part of the manuscript is very brief on purpose as the authors felt like it was unnecessarily long in the first place and we decided to refer to the methods used nevertheless. We have added the missing reference to the NMR method used in this research.
The reference from 1991 is a primary literature reference and it was chosen to acknowledge the original work. This reference clearly contains valuable information related to work about total carbon and nitrogen characteristics in proteins, not the methodology itself. Otherwise, we would like to keep the references already present in the manuscript in order to cover more literature, since MDPI does not impose a reference count limitation.

Reviewer 2 Report
The subject of the article was the analysis of the “determination of floral origin markers of Latvian honey by using IRMS, UHPLC-HRMS, 1H-NMR”. The measurements were carried out correctly, all the necessary parameters were determined and checked. The text is written in correct English, the structure is simple and clear. However, the article has many shortcomings and needs to be improved. Therefore, I recommend major revision.
I am asking the authors to respond and make appropriate changes to the text, to the following comments:
- The authors examined a large pool of samples using a variety of research methods. However, out of 78 samples, most of them 55 are multiflorous plants, only a few samples are monofloral: 4 buckwheat, 6 clover, 3 heather, 3 linden, 4 rape, 3 willow. In my opinion, three samples of a given honey are not enough for statistical considerations, especially in comparison with 55 multiflorous.
- UHPLC-HRMS analysis for polyflorals showed the multiflorous samples have a large number of the compound content (31 substances). However, only the content of a few compounds turned out to be different for mono- and multiflorous honeys. The data spreads for multiflorous samples are very large and the number of monoflowering samples is very small. Thus, the results are nevertheless of little diagnostic value (maybe with the exception of rutin for buckwheat as shown by PCA analysis).
- In the introduction, the authors mention the problem of counterfeit honey. It is a pity that there were no such samples. Do the authors have any data on this subject? If not, it is worth comparing data published already.
- Honeys are adulterated by sugar addition. Why did the authors not investigate the content and types of sugars in the samples?
- NMR analysis - it is a pity that the authors did not develop this section. The qNMR research would be very interesting.
- Other:
- Line 251 – “polyfloral honey contained 53 ± 126 μg/kg”. content less than zero?
- Line 251-252 – “It was less found in linden and rapeseed honey but not found at all in heather honey.” – why the content in was not marked "not detected"? There is a point for heather with high uncertainty in the Fig 2a.
- Figure S7 – chromatograms overlap and are unreadable
Author Response
REVIEWER 2
Comments and Suggestions for Authors
The subject of the article was the analysis of the “determination of floral origin markers of Latvian honey by using IRMS, UHPLC-HRMS, 1H-NMR”. The measurements were carried out correctly, all the necessary parameters were determined and checked. The text is written in correct English, the structure is simple and clear. However, the article has many shortcomings and needs to be improved. Therefore, I recommend major revision.
I am asking the authors to respond and make appropriate changes to the text, to the following comments:
- The authors examined a large pool of samples using a variety of research methods. However, out of 78 samples, most of them 55 are multiflorous plants, only a few samples are monofloral: 4 buckwheat, 6 clover, 3 heather, 3 linden, 4 rape, 3 willow. In my opinion, three samples of a given honey are not enough for statistical considerations, especially in comparison with 55 multiflorous.
Response:
We appreciate the reviewer’s comment and we completely agree that the sample pool is quite small. All of the samples were initially collected directly from the beekeepers as a part of this project and were declared as monofloral honey samples by the beekeepers themselves. However, the melissopalynology analysis proved otherwise and thus we were left with few “true” monofloral variety honeys that would reach a high enough specific pollen threshold. A part of the truly monofloral samples were collected directly from the Beekeepers Association. Nevertheless, a proof-of-concept study was needed with the available resources in order to highlight the possibility of further research.
We have now added additional text to the manuscript 2.1. Samples section to better highlight our choice of samples:
In manuscript lines 85-88 text has been changed from:
“A total of 78 different honey samples were collected in the territory of Latvia and were classified as natural origins. Botanical origins of samples were approved by melissopalynology analysis [17].”
“A total of 78 different honey samples were collected directly from the beekeepers in the territory of Latvia, declared as of natural origin and of specific monofloral varieties. The true botanical origin of the samples was further examined by melissopalynology analysis [17] and later confirmed or deemed of lesser, polyfloral quality.”
- UHPLC-HRMS analysis for polyflorals showed the multiflorous samples have a large number of the compound content (31 substances). However, only the content of a few compounds turned out to be different for mono- and multiflorous honeys. The data spreads for multiflorous samples are very large and the number of monoflowering samples is very small. Thus, the results are nevertheless of little diagnostic value (maybe with the exception of rutin for buckwheat as shown by PCA analysis).
Response:
Yes, rutin is the most suitable marker for buckwheat honeys. Nonetheless, other compounds than rutin were investigated just for the purpose of exploring the variation within the data set and for the sake of reporting the results.
The reason for the huge spread is that in a lot of samples there were high levels of pollen that did not meet the monofloral threshold and thus spread the data. We have tried to explain how some of specific pollen types and corresponding compounds could attribute to the spread towards floral class clusters, e.g., “This comes in good agreement with melissopalynology results because buckwheat (Fagopyrum esculentum) pollens were found in clover and willow monofloral honeys in a range of 0-6%.” and in lines 312-318. In hindsight, a pooled quality control sample would have been a great choice to avoid such a data spread, but it is near to impossible to incorporate at this stage of the project.
The larger sample size of polyfloral was kept in order to strongly indicate characteristic concentration levels while the low sample size of monofloral samples, indeed, just shows approximative characteristics.
Additionally, a targeted analysis method with 31 substances used for quantitative analysis would not allow us to find suitable markers for individual floral classes, hence, only rutin was found as most suitable. Non-targeted analysis methods would allow for the identification of additional marker substances, which, in hand with targeted method development, could expand the capabilities of this proof of concept study.
- In the introduction, the authors mention the problem of counterfeit honey. It is a pity that there were no such samples. Do the authors have any data on this subject? If not, it is worth comparing data published already.
Response:
We wanted to narrow down the subject of honey quality analysis. Counterfeits are a truly intriguing subject and, yes, we have done C4 sugar tests in a level of preliminary investigations, but have not reported the results. However, we have never encountered this specific type of adulteration. Also, in the context of Latvia, adulteration is a lesser concern rather than the variety of floral origins confirmation (and thus avoidance of false-labelling).
We mentioned the problem of counterfeits because it is a global-scale problem and attention to it must be deserved. Perhaps (to be honest, most likely) in the future there will be a flood of honey imports from third world countries which might be of lesser quality or even adulterated. This is why in recent years our institutions have been fingerprinting local honey using various analytical methods in order to later identify the counterfeit produce which might be labelled as locally manufactured.
- Honeys are adulterated by sugar addition. Why did the authors not investigate the content and types of sugars in the samples?
Response:
Since the focus of our study was on the false-labelling of floral origins, we decided to pay less attention to sugars and, therefore, investigate minor compounds like polyphenols or the 1H-NMR profiles. We have preliminary results for sugar disbalance after adulteration but this was not considered to fit in for this study. Also, the samples are not obtained via commercial source but obtained directly by the collaboration of beekeepers who confirm that honey is of natural origin.
- NMR analysis - it is a pity that the authors did not develop this section. The qNMR research would be very interesting.
Response:
The authors appreciate the reviewer's comment. We completely agree that qNMR method development would be really nice, but we were limited to qualitative 1H-NMR analysis that is commonly used for the discrimination of various food or biological samples and has proven to work well in conjuncture with other analytical methods.
We believe that qNMR might have a high potential for honey characterization but this analysis method needs to be carefully optimised and validated due to the low concentration of specific analytes in the monofloral honey samples and the overall complexity of honey sample matrix.
- Other:
- Line 251 – “polyfloral honey contained 53 ± 126 μg/kg”. content less than zero?
Response:
The text in line 251 has been edited to describe a range of concentrations rather than a mean with SD: “contained from < 5 (LOQ) to 696 μg/kg with a mean of 53 μg/kg”
- Line 251-252 – “It was less found in linden and rapeseed honey but not found at all in heather honey.” – why the content in was not marked "not detected"? There is a point for heather with high uncertainty in the Fig 2a.
Response:
The image was generated using ANOVA one-way analysis tool from Minitab software. The heather sample group was kept to see if the difference between other sample groups is significant even though the heather honey sample size is small and does not contain any values (basically it is the LOQ value). The generated image contained a heather “0 value” point with confidence intervals which in case of lack of an overlay indicates statistically different means. Since the number of monoflorals is low, we wanted to be sure that means are truly different even in this scenario.
Fig 2a design has been edited to be more unambiguous since the rutin was not found in monofloral samples of heather honey and confidence intervals do not interfere with buckwheat honey concentration confidence intervals.
- Figure S7 – chromatograms overlap and are unreadable
Response:
Text in corresponding lines has been edited and Figure S7 has been improved visually and it now has a comprehensive overview of 1H-NMR spectra of different monofloral types.

Round 2
Reviewer 1 Report
Please correct "investigate a methodology" with "use of different methodologies". See line 75.
Author Response
REVIEWER 1
Comments and Suggestions for Authors
- Please correct "investigate a methodology" with "use of different methodologies". See line 75.
Response:
Authors have taken into account the comment and sentence in line 75 is edited as follows:
"The main aim of this study was the use of different methodologies to classify the botanical origin of various types of monofloral Latvian honey to target the mislabeling of protected destination of origin (PDO) products. "
We appreciate your invested time and useful comments about vital parts of the article.
Reviewer 2 Report
The manuscript has been sufficiently corrected. I have no further comments on the text.
Author Response
REVIEWER 2
Comments and Suggestions for Authors
- The manuscript has been sufficiently corrected. I have no further comments on the text.
Response:
Autors feel grateful for your given comments and suggestions, which notably have improved articles quality.